# High-Speed and High-Power Ferroelectric Switching Current Measurement Instrument for Materials with Large Coercive Voltage and Remanent Polarization

**DOI:** 10.3390/s22249659

**Published:** 2022-12-09

**Authors:** Keisuke Yazawa, Andriy Zakutayev, Geoff L. Brennecka

**Affiliations:** 1Department of Metallurgical and Materials Engineering, Colorado School of Mines, Golden, CO 80401, USA; 2Materials Science Center, National Renewable Energy Laboratory, Golden, CO 80401, USA

**Keywords:** ferroelectrics, wurtzite, nitride, switching current detection, KAI model, NLS model

## Abstract

A high-speed and high-power current measurement instrument is described for measuring rapid switching of ferroelectric samples with large spontaneous polarization and coercive field. Instrument capabilities (±200 V, 200 mA, and 200 ns order response) are validated with a LiTaO_3_ single crystal whose switching kinetics are well known. The new instrument described here enables measurements that are not possible using existing commercial measurement systems, including the observation of ferroelectric switching in large coercive field and large spontaneous polarization Al_0.7_Sc_0.3_N thin films.

## 1. Introduction

Ferroelectric switching dynamics have been of interest since the emergence of ferroelectric materials [1,2,3,4,5]. Kolmogorov–Avrami–Ishibashi (KAI) and nucleation limited switching (NLS) models have been developed to explain the phenomena based on experimentally measured polarization evolution [2,4]. Switching polarization is quantified by measuring charge flow in external circuits. The classic Sawyer–Tower circuit stores and reads the charge on a reference capacitor, essentially integrating the switching current over time [6]. Camlibel reported a voltage transient attributed to polarization switching across a reference capacitor following voltage pulse application in ms order using the Sawyer–Tower circuit [7]. However, to obtain the switching response in < µs order, a direct measurement of switching current flowing through a circuit is suitable for time-resolved observation of current/polarization dynamics. Measuring the voltage drop across a simple shunt resistor due to the flowing current introduces a tradeoff between measurable signal and circuit load effects, which are particularly problematic for the rapid dynamic change in impedance and the dynamic current range seen in ferroelectric switching [8]. Thus, a transimpedance amplifier is frequently used because its virtual ground avoids circuit load effects across the wide impedance changes inherent to devices such as photodiodes and switching ferroelectrics [9,10,11]. Observation of sample-limited ferroelectric switching behavior requires integration of an appropriate voltage source in addition to time-resolved charge measurement.

The emerging wurtzite ferroelectrics [12] exacerbate such measurement challenges because they exhibit massive switchable polarization values (>100 µC/cm^2^) and very square loops, which translate into large switching currents, as well as large coercive fields combined with degraded film quality at very thin layers [13], which together necessitate relatively large voltages for switching. For example, Al_0.7_Sc_0.3_N typically exhibits coercive fields of 5000 kV cm^−1^, which requires 150 V for a film of 300 nm thickness [12,14]. Maximum switching current can be estimated under the assumption of triangular current peak, which is
$$i_{s}=\frac{P_{s}A}{t_{FWHM}}$$
where *P_s_* is the spontaneous polarization, *A* is the capacitor size, and *t_FWHM_* is the current peak width. The switching current of a 100 µm diameter capacitor of Al_0.7_Sc_0.3_N (*P_s_* = 110 µC cm^−2^ [14,15,16,17,18]) reaches approximately 100 mA for a 100 ns switching time. Thus, a measurement system capable of measuring the intrinsic behavior of the sample without the circuit itself dominating the measurement requires a total design including large current and voltage capabilities. Commercially available current measurement systems are usually limited by either the speed or voltage; a high speed pulse I–V system has a lower voltage range (±40 V) [19] and a high voltage instrument shows a slow slew rate (~0.5 V/μs) [20].

In this study, we report the development and demonstration of a high speed, high power current measurement instrument capable of sub-microsecond measurements of ferroelectric switching even for high coercive voltage (*V_c_*) and high *P_s_* thin film samples, which stands in the gap between commercially available high speed and high voltage current measurement systems. The ±200 V, 200 mA instrument demonstrates sample-controlled current without interference of the external circuit down to at least 200 ns. After validation using a well-characterized 60 µm thick LiTaO_3_ single crystal, the instrument is used to measure anomalous switching kinetics of a 250 nm thick Al_0.7_Sc_0.3_N film, characteristics that would remain overlooked without such measurement capabilities.

## 2. Measurement Instrument

Figure 1 shows the schematic of the measurement instrument described here. A voltage sequence coming from the function generator is amplified by a commercially available high voltage/high speed voltage amplifier (maximum output: ±200 V and 200 mA, cutoff frequency: 1.2 MHz). This signal goes to both the sample and the high impedance input of an oscilloscope for reference. The voltage applied to the sample drives the ferroelectric switching, and current flows into the virtually grounded transimpedance amplifier input. The current is converted to voltage in the transimpedance amplifier and measured at a high rate by the oscilloscope.

The feedback circuit constants *R_f_* and *C_f_* in the transimpedance amplifier are determined by initial estimations of the switching current and capacitance of the saturated ferroelectric. Based on the spontaneous polarization and relative permittivity, the switching current for a 50 µm diameter capacitor of Al_0.7_Sc_0.3_N is estimated to be roughly 20 mA at 100 ns, and the capacitance is roughly 1 pF [12,14,21]. Thus, given the output voltage range (±5 V) of the operational amplifier (OPA657), *R_f_* is set to 47 Ω. The *C_f_* is set to >5 pF to compensate the phase and create phase margin to stabilize the operation amplifier and avoid oscillation [22]. Based on these values and the gain bandwidth product of the operational amplifier, the cutoff frequency of the transimpedance amplifier is calculated to be <950 MHz at maximum without any parasitic component consideration [22], which is much faster than that of the voltage amplifier, so that current detection is not limited by the transimpedance amplifier. Therefore, the cutoff frequency of the voltage amplifier (1.2 MHz) represents the maximum frequency (minimum time) capability of the circuit, and measurements at slower speeds will be dominated by the sample response rather than the circuit capabilities.

## 3. Materials

Circuit capabilities were verified via measurements on samples exhibiting well-known switching characteristics. Figure 2 shows ferroelectric P–E loops measured for a commercially available 60 µm thick vapor transport equilibrated (VTE) LiTaO_3_ single crystal with a top contact diameter of 100 µm (Figure 2a) and a 250 nm thick Al_0.7_Sc_0.3_N film with a 50 µm diameter top contact (Figure 2b). The polycrystalline textured Al_0.7_Sc_0.3_N film is deposited via radio frequency reactive sputtering method on a Pt/TiO_x_/SiO_2_/Si substrate (the details of the deposition condition can be found elsewhere [14]). Both exhibit robust ferroelectricity with a coercive voltage of 63 V for LiTaO_3_ and 110 V for Al_0.7_Sc_0.3_N under a 10 kHz triangular excitation voltage. The LiTaO_3_ loop is well saturated as the sample exhibits a high resistance after switching, whereas the much thinner Al_0.7_Sc_0.3_N sample shows evidence of leakage current contribution to the hysteresis loop. Thus, a positive-up-negative-down (PUND) method is employed to subtract the leakage current contribution [23,24].

## 4. Results and Discussion

Typical PUND measurement results for the Al_0.7_Sc_0.3_N thin film using this instrument are shown in Figure 3. Along with the applied voltage sequence (black line), the current response (red line) is shown in Figure 3a. The significant positive current (14 mA) and negative current (−27 mA) peaks associated with the P and N pulses correspond to polarization reversal (ferroelectric switching) within the sample. The asymmetric responses point to the imprint seen in the P–E loops in Figure 2b and reported previously [14]. Figure 3b,c shows the magnified current and voltage evolution for the P and U pulses. For both pulses, capacitive current is clearly measured during the voltage rise, confirming that the transimpedance amplifier does not limit the current flow during this stage as designed in the Measurement Instrument section. Note that the CR time constant is orders of magnitude faster than the applied voltage rise time due to the pF order of capacitance of the measured devices. A large current peak appears after the capacitive current only for the switching pulses (P and N). This switching current is not disturbed by the external circuit; the current is below the current supply limit (200 mA) of the voltage amplifier, and no voltage drop associated with power shortage at the current peak is seen in the measured applied voltage curve in Figure 3b.

Switching current is determined by subtracting the non-switching (capacitive plus leakage) currents of the U (or D) cycles from the large (switching plus capacitive plus leakage) currents of the P (or N) cycles. Switched polarization as a function of time, *P*(*t*), is simply expressed as
$$P_{\left(t\right)}=\frac{1}{A}\int \left(i_{p,n\left(t\right)}-i_{u,d\left(t\right)}\right)dt$$
where *A* is the capacitor area, *i_p,n_*_(*t*)_ is the current of the P or N pulse, and *i_u,d_*_(*t*)_ is the current during the U or D pulse. For a complete switching event, the total *P*(*t*) equals 2P_r_.

Figure 4 shows the time-resolved polarization evolution of the LiTaO_3_ single crystal and Al_0.7_Sc_0.3_N thin film samples studied here. Voltage rise time is often defined as either the time required for the voltage value to go from 0–100% of its max or from 10–90%. In order to define *t* = 0 for our polarization–time plots, we use these two definitions of rise time as uncertainty bounds, as shown in Figure 4a: *t_r_*_1_ = 200 ns for when voltage reaches 90% of amplitude and *t_r_*_2_ = 330 ns for when voltage reaches 100% of amplitude. These values are reasonable based on the voltage amplifier cutoff frequency *f_c_* = 1.2 MHz and the rise time–cutoff frequency relationship *t_r_* = 0.35/*f_c_* [25]. Figure 4b shows the polarization evolution curves for the LiTaO_3_ single crystal for maximum applied voltages between 60 and 200 V. In this figure, *t* = 0 is defined as the applied voltage pulse starting time at V = 0. Sample polarization reversal both starts sooner and occurs more quickly as the applied voltage increases; for an applied voltage of 200 V, the polarization evolution starts between t_r1_ and t_r2_, indicating that the circuit capabilities contribute meaningful uncertainty to the measurement at that point. Hereinafter, *t* = 0 is defined using both *t_r_*_1_ and *t_r_*_2_ as bounds represented by error bars. Figure 4c shows polarization evolution curves for the LiTaO_3_ sample with a 130 ns uncertainty range for *t* = 0 as redefined using *t_r_*_1_ and *t_r_*_2_.

The polarization evolution curves in Figure 4c can be described using the classic KAI model [2] and are consistent with prior analysis of LiTaO_3_ switching dynamics [5]. The KAI model is expressed as
$$f=1-\exp\left\{-{\left(\frac{t}{t_{0}}\right)}^{n}\right\}$$
where *t*_0_ is the characteristic time and *n* is the Avrami exponent that represents the curve slope. The Avrami exponents for the curves are *n* = 2 for all different voltages, as shown in Figure 4d. This indicates two-dimensional domain growth and coalescence, which is commonly seen in thin films, polished specimens, and single crystals where nucleation is essentially limited to specimen surfaces [26]. This result further validates both the measurement circuit capabilities and the definition of *t* = 0 based on the voltage rise time.

Polarization evolution curves for the Al_0.7_Sc_0.3_N film are shown in Figure 4e, for voltages from 105 to 125 V. The applied voltage is limited to the sample breakdown occurred >130 V. Similar to LiTaO_3_, the curves shift to shorter times with higher voltage applied. However, unlike LiTaO_3_, the slopes of the curves increase for higher voltages as well. Figure 4f shows the Avrami exponent as a function of the applied voltage. At 105 V, the Avrami exponent is 2, consistent with two-dimensional domain growth and coalescence, which is reasonable for a thin film. This result is in good agreement with slower switching dynamic measurements (>1 ms) reported by Fichtner on similar films [15]. At 125 V, however, the Avrami exponent reaches 5–7 (range due to the uncertainty of *t* = 0), which is beyond the classic *n* = 1 + *D* interpretation of the KAI model that assumes D is the dimensionality of kinetically limited domain growth and coalescence. The NLS model of ferroelectric switching kinetics is also not directly applicable because it rationalizes smaller Avrami exponents *n* < 1 based upon kinetics limited by only nucleation without growth and coalescence. Thus, the large Avrami exponents measured here imply that a model beyond the classic KAI and NLS models is required to explain the switching of Al_0.7_Sc_0.3_N ferroelectrics.

## 5. Conclusions

In summary, a high-speed and high-power current measurement instrument is developed for measuring the rapid polarization reversal of ferroelectric samples that exhibit large spontaneous polarization values and high coercive fields. The operation of this instrument is validated using a LiTaO_3_ single crystal with well-described switching characteristics. The KAI model appropriately describes the switching of LiTaO_3_ with an Avrami exponent of *n* = 2 up to 200 V across a 60 µm thick crystal for applied voltage rise times down to 100 ns. The results validate the instrument capability (±200 V and 200 mA response time on the order of 200 ns). The instrument use is then extended to the measurement of polarization reversal in an Al_0.7_Sc_0.3_N thin film that also shows robust ferroelectric hysteresis loops and switching behavior at low speeds, consistent with prior reports. Measurements of the rapid (<1 µs) switching behavior of Al_0.7_Sc_0.3_N enabled by this new measurement instrument reveal non-physical switching dynamics with the classic KAI and NLS models. The instrument developed and validated here enables measurement and an improved understanding of the switching kinetics of the growing family of large coercive field square-loop ferroelectrics.

## Figures and Tables

**Figure 1 sensors-22-09659-f001:**
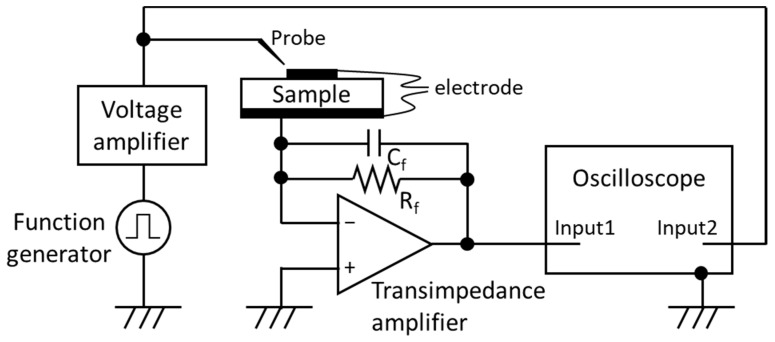
Schematic of the measurement instrument.

**Figure 2 sensors-22-09659-f002:**
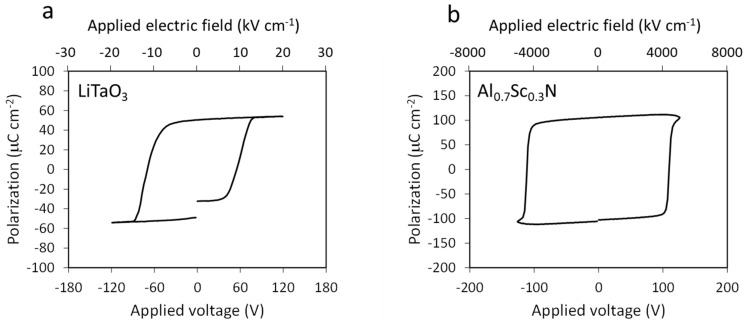
Ferroelectric hysteresis loops for measured samples: (**a**) 60 µm thick LiTaO_3_ single crystal and (**b**) 250 nm thick Al_0.7_Sc_0.3_N film.

**Figure 3 sensors-22-09659-f003:**
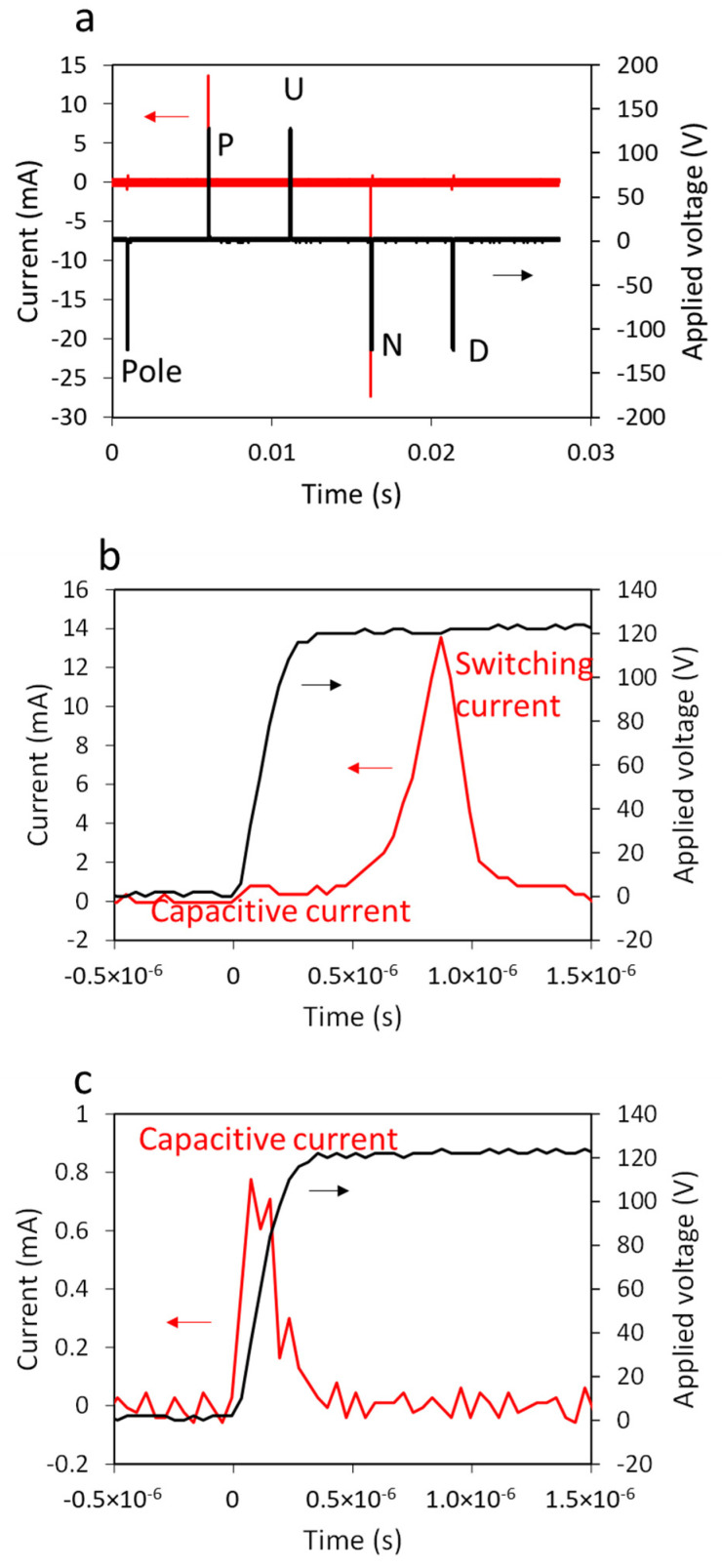
Typical PUND measurement results for Al_0.7_Sc_0.3_N. (**a**) Full sequence of applied voltage and measured current. (**b**) Magnified applied voltage and current curves during P pulse. (**c**) Magnified applied voltage and current curves during U pulse. Red lines represent measured current (red arrows), and black lines represent applied voltage (black arrows).

**Figure 4 sensors-22-09659-f004:**
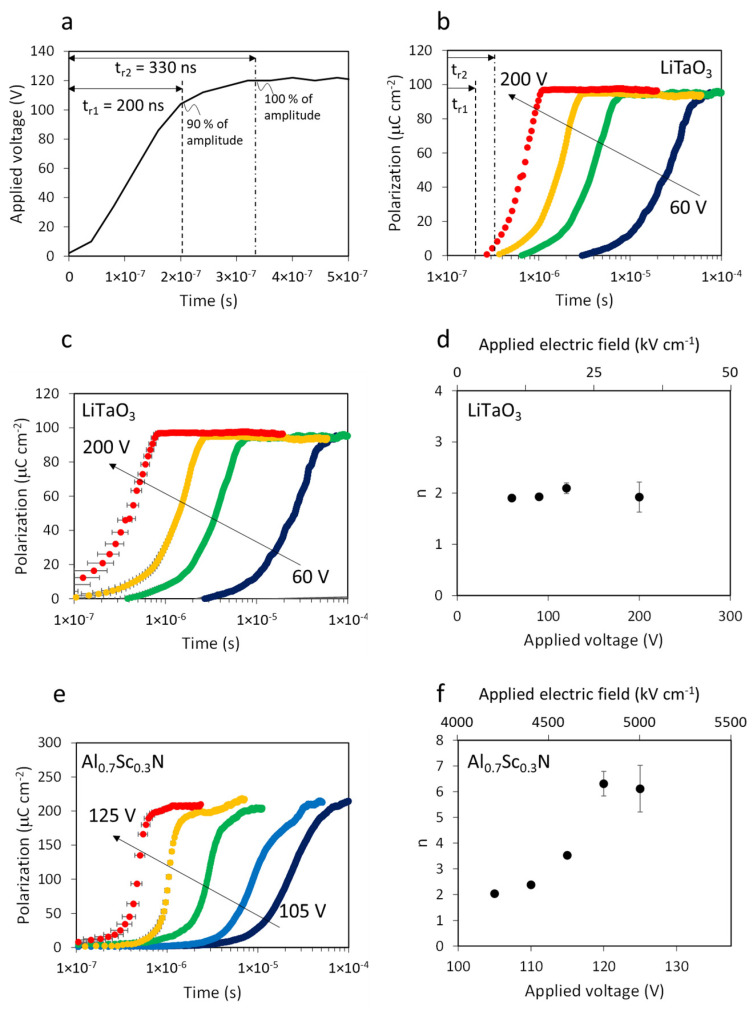
Polarization evolutions based on PUND current measurement. (**a**) Applied voltage pulse rise time analysis to define *t* = 0. (**b**) Polarization evolution for LiTaO_3_ with various voltages. Defined *t* = 0 is the voltage pulse starting time. (**c**) Polarization evolution for LiTaO_3_ with various voltages. Defined *t* = 0 is t_r1_ and t_r2_ shown in (**a**). (**d**) Avrami exponents from polarization evolution curve of LiTaO_3_ (**e**) Polarization evolution for Al_0.7_Sc_0.3_N with various voltages. Defined *t* = 0 is *t_r_*_1_ and *t_r_*_2_ shown in (**a**). (**f**) Avrami exponents from polarization evolution curve of Al_0.7_Sc_0.3_N.

## Data Availability

The data affiliated with this study are available from the corresponding author upon reasonable request. The views expressed in the article do not necessarily represent the views of the DOE or the U.S. Government.

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
