# Peer review of "High-Speed and High-Power Ferroelectric Switching Current Measurement Instrument for Materials with Large Coercive Voltage and Remanent Polarization"

_sensors, 2022, doi:10.3390/s22249659_

Round 1

Reviewer 1 Report

Keisuke Yazawa et al. reported a superior ferroelectric switching behavior using the state-of-the-art measurement instrument. The development of precision instrument strongly boosts the collection of ferroelectric signals under a high-speed and high-power condition. This work provides an important indicator for monitoring ferroelectric switching dynamics in those materials with large coercive voltage and remnant polarization. I recommend this manuscript for publication in this journal until the following issues are addressed.

1. What is the key in your measurement system to realize related function characteristic? Or what is the core difference between your instrument and exsiting commercial measurement system.

2. Polarizing LiTaO3 does not need too large electric field. Are the related measuring results the same as the commercial instrument?

3. In the last paragraph in the introduction section, the unit of thickness of LiTaO3 is missing.

4. Can the authors give some explanations about the deviations from KAI switching in Al0.7Sc0.3N films?

Reviewer 2 Report

The authors designed a high-speed and high-power current measurement instrument for the measurement of rapid switching of ferroelectric samples with large spontaneous polarization and coercive field. The designed instrument show excellent performance. I suggest this manuscript be accepted in present form.
